# Smoking is associated with higher risk of contracting bacterial infection and pneumonia, intensive care unit admission and death

**Karl Stattin**[1]*, **Mikael Eriksson**[1], **Robert Frithiof**[1], **Rafael Kawati**[1], **Douglas Crockett**[2], **Michael Hultström**[1,3], **Miklos Lipcsey**[1,4]

**1** Department of Surgical Sciences, Anesthesiology and Intensive Care, Uppsala University, Uppsala, Sweden, **2** Nuffield Department of Clinical Neurosciences, University of Oxford, Oxford, United Kingdom, **3** Department of Medical Cell Biology, Integrative Physiology, Uppsala University, Uppsala, Sweden, **4** Hedenstierna Laboratory, Department of Surgical Sciences, Uppsala University, Uppsala, Sweden

* Karl.stattin@uu.se

## Abstract

**Data Availability Statement:** Participant consent does not support the public sharing of data, but all data are available from the SIMPLER steering

### Background

Smoking has been associated with a higher risk of contracting pneumonia, but contradictory results have shown that smoking may or may not *decrease* the risk of dying in pneumonia. The aim of this study is to investigate how smoking is associated with contracting any infection and pneumonia and death.

### Method and findings

Participants were drawn from the population-based Cohort of Swedish Men and the Swedish Mammography Cohort, which are representative of the Swedish population. Participants have answered detailed lifestyle questionnaires and have been followed in national registers, such as the Patient Register, Cause of Death register and Swedish Intensive Care Registry. The risks of contracting infection and pneumonia or dying in infection and pneumonia were assessed using Cox regression. Of 62,902 cohort participants, 25,297 contracted an infection of which 4,505 died; and 10,471 contracted pneumonia of which 2,851 died. Compared to never smokers, former smokers at baseline had hazard ratio (HR) 1.08 (95% confidence interval (CI) 1.05–1.12) of contracting and HR 1.19 (95% CI 1.11–1.28) of dying in infection and HR 1.17 (95% CI 1.12–1.23) of contracting and HR 1.16 (95% CI 1.06–1.27) of dying in pneumonia during follow-up. Compared to never smokers, current smokers at baseline had HR 1.17 (95% CI 1.13–1.21) of contracting infection and HR 1.64 (95% CI 1.52–1.77) dying in infection; HR 1.42 (95% CI 1.35–1.49) of contracting pneumonia and HR 1.70 (95% CI 1.55–1.87) of dying in pneumonia during follow-up. The risk of contracting and dying in infection and pneumonia increased in a dose-response manner with number of pack years smoked and decreased with years since smoking cessation.

committee (at www.simpler4health.se) to researchers with ethical approval

**Funding:** The author(s) received no specific funding for this work.

**Competing interests:** The authors have declared that no competing interests exist.

## Conclusion

Smoking is associated with contracting and dying in any infection and pneumonia and the risk increases with pack years smoked, highlighting the importance of both primary prevention and smoking cessation.

## Introduction

Although an established risk factor for numerous diseases [1, 2], smoking remains one of the leading preventable causes of death in the world [3] and an estimated 1.1 billion people smoke globally [3]. The consequences of smoking is unevenly studied. Whereas smoking is a recognised risk factor for e.g. lung cancer [4] and chronic obstructive pulmonary disease [5], less is known about its association with bacterial infections.

Pneumonia is one of the most common bacterial infections [6], and although smoking has repeatedly been shown to increase the risk of pneumonia [7–10], the association between smoking and death in pneumonia is contentious. Some studies indicate a higher risk of death in pneumonia among current smokers [11, 12], while other studies show no association [11, 13, 14] or even indicate a *lower* risk of death [15–17]. This discrepancy warrants further investigation. Smokers contracting pneumonia are younger and have fewer comorbidities but higher alcohol consumption than non-smokers suffering from pneumonia [12], which may affect the outcome. Although the deleterious effects on the immune system caused by smoking extend beyond the respiratory tract [18], the association between smoking and non-respiratory bacterial infections has not been widely studied, and whether smoking increases the risk of dying in infection and sepsis is unclear [19–23].

Thus, the associations between smoking and infection generally and pneumonia specifically are incompletely understood. The aim of the present study is therefore to investigate the associations between smoking and contracting any infection and pneumonia, intensive care unit admission and death, adjusted for confounders using large representative population-based cohorts.

## Method

### Participants

The Cohort of Swedish Men (COSM) was started by mailing all men living in Västmanland and Örebro counties born between 1918–1952 a lifestyle questionnaire in 1997, to which 49% responded (n = 48,850). The Swedish Mammography Cohort (SMC) invited all women living in Uppsala county born between 1914–1948 and all women living in Västmanland county born between 1917–1948 to answer a lifestyle questionnaire in 1987 with a response rate of 74% (n = 66,651). A second questionnaire was administered to the women in SMC in 1997 with a response rate of 70% (n = 39,984). Data collection commenced on September 15th 1997 and ended on December 31st 1997 for both cohorts, and participants were followed from January 1st 1998. Both cohorts are representative of the Swedish population [24, 25] and have been followed in national registers since inception. See cohort flowchart in S1 Fig.

### Exposure and confounders

Cigarette smoking was ascertained in the 1997 lifestyle questionnaire through questions enquiring about ever regularly smoking cigarettes; if the participant had stopped smoking, and

if such, how many years ago; the number of cigarettes smoked per day at the age of 15–20, 21–30, 31–40, 41–50 and 51–60 years, respectively; as well as the number of cigarettes smoked per day this year. From these questions, smoking status (defined as never, former or current smoking at baseline in 1997), number of pack years smoked and years since smoking cessation were derived.

Confounders were selected using Directed Acyclic Graphs (DAG) [26] and were collected from questionnaires in the case of demographic or lifestyle covariates, and from the National Patient Register in the case of comorbidities to construct Charlson's weighted comorbidity index [27].

## Outcome

Outcomes of interest were contracting any infection or pneumonia, intensive care unit (ICU) admission due to infection or pneumonia and death in infection or pneumonia. Diagnosis of infection was collected from the National Patient Register, which has high validity and near-complete coverage [28]. ICU admission, physiological derangement at admission (estimated using Simplified Acute Physiology Score 3, SAPS3 [29]), organ supportive treatment and ICU length of stay were collected from the Swedish Intensive Care Registry [30]. Cause of death was acquired from the Cause of Death Register. For the main analysis, deaths were counted both if infection was judged as the underlying cause of death or as a contributing cause of death.

For any infection, International Statistical Classification of Diseases and Related Health Problems (ICD-10) codes of A15, A16, A17, A18, A19, A32, A39, A40, A41, A46, A48, A49, B95, B96 D65, G00, G01, G02, G03.9, G05.0, G06, G07.9, I33, I39, J13, J14, J15, J16, J18, J85, J86, K35, K57.0, K57.2, K57.4, K57.8, K63.0, K63.1, K65, K80.0, K80.1, K80.3, K80.4, K81, K83.0, K85, M00, M01, M72.6, N10, N12, N13.6, N39.0, N70, R65.1, R57 and T802 were compiled from the registers. For pneumonia, ICD-10 codes J13, J14, J15, J16, J18, J85 and J86 were used.

## Statistical analyses

Data was accessed for analysis from November 21$^{st}$ 2023 to March 1$^{st}$ 2024. Hazard ratio (HR) and corresponding 95% confidence intervals (CI) were calculated using Cox proportional hazards regression with attained age as time scale. Cohort participants contributed time at risk from January 1$^{st}$ 1998 until the outcome of interest, death, or end of follow up at December 31$^{st}$ 2021, whichever occurred first. Separate analyses were performed for any infection and pneumonia, for the outcomes of contracting disease, for intensive care unit admission and for death. The exposures of interest were smoking status at baseline in 1997 (never/former/current), number of pack years smoked (never smoker/<10/10-20/20-30/30-40/>40) and years since smoking cessation (never smoker/<10/10-20/20-30/>30), which were first examined in a crude model including sex (man/woman), and subsequently in an adjusted model including sex, Charlson's weighted comorbidity index (continuous), education (<9 years/9-12 years/>12 years/other or vocational), marital status (living alone, i.e. single, divorced, widow or widower versus cohabiting, i.e. married or living with someone), hours of weekly exercise (<1/1/2-3/4-5/>5), minutes of daily walking or bicycling (hardly ever/<20 /20-40/40-60/>60) and alcohol consumption (grams per day, continuous). In the fully adjusted model, the analysis of years since smoking cessation was further adjusted for pack years (continuous). The assumption of proportional hazards was visually assessed using log-log plots.

The proportion of missing data were 11.0% for exercise, 9.1% for walking, 8.4% for alcohol consumption, 6.6% for marital status and 1.8% for smoking status. Missing data for all other

variables were <1%. Outcome variables did not have missing information. Complete-case analysis was performed with 62,902 individuals having all prerequisite data to be included in the adjusted analysis.

To adjust for frailty, a sensitivity analysis was performed adjusted for self-rated health (very good/good/neither good nor bad/bad/very bad). To account for former smokers that may have quit due to illness, a sensitivity analysis was performed with follow-up beginning three years after baseline. To test the robustness of the analysis of death, a sensitivity analysis was performed where only cases when infection was considered the underlying cause of death were included. A sensitivity analysis was performed further adjusting for body mass index (BMI, i.e. weight in kilograms divided by length in meters squared), divided into categories: <20/20-25/25-30/>30. A sensitivity analysis was performed with time since 1998 as the time scale in Cox proportional hazards, adjusted for age at baseline as a continuous variable. To test if smokers have lower risk of death in *one specific episode* of pneumonia, a sensitivity analysis was performed investigating 30-day mortality after the first episode of pneumonia.

The study was approved by the National Ethical Review Agency, Stockholm, Sweden (DNR 2022-05751-01) and the Declaration of Helsinki was followed. The STROBE statement was followed for reporting. All participants gave written informed consent for inclusion. Participant consent does not support the public sharing of data, but all data are available from the SIMPLER steering committee (at www.simpler4health.se) to researchers with ethical approval. All analyses were performed on Stata 15.1 (Stata Corp., College station, Texas, United States of America). Descriptive data are presented as median (interquartile range, IQR) and number (%).

## Results

Of cohort participants, 26,138 (41.6%) were women. Participants median age at baseline was 60 years (IQR, 53–68), 26,377 (41.9%) were never smokers, 21,405 (34.0%) former smokers and 15,121 (24.0%) current smokers in 1997 (Table 1). Among ever smokers, the median number of pack years smoked was 16 years (IQR 8–27). Among participants having quit smoking, the median time since smoking cessation was 18 years (IQR 10–26). Among 1,049 participants admitted to an ICU due to infection during the study period, 247 (23.6%) were women; the median age at admission was 76 years (IQR 71–81); 330 (31.5%) were never smokers, 381 (36.3%) former smokers, and 338 (32.2%) current smokers at baseline in 1997; the median SAPS3 was 67 (IQR 59–76); and 379 (36.1%) were treated with invasive mechanical ventilation (S1 Table).

### Any infection

During follow-up, 62,902 individuals contributed 1,228,307 person-years at risk (PYAR), during which 25,297 individuals suffered any infection, 1,049 were admitted to an ICU due to infection and 4,505 died from infection. Compared to never smokers, participants reporting former smoking in 1997 had HR 1.08 (95% CI 1.05–1.12) of contracting an infection, HR 1.21 (95% CI 1.04–1.41) of being admitted to an ICU and HR 1.19 (95% CI 1.11–1.28) of dying in infection. Compared to never smokers, the risks for participants reporting current smoking in 1997 were HR 1.17 (95% CI 1.13–1.21); HR 1.64 (95% CI 1.41–1.92) and HR 1.64 (95% CI 1.52–1.77) respectively (Fig 1 and S2 Table). Pack years smoked had a dose-response association with a higher risk of contracting infection, ICU admission and dying. Years since smoking cessation was negatively associated with contracting infection, ICU admission and dying with a weak dose-response pattern (Fig 2). Sensitivity analyses showed similar results as the main analysis when adjusted for self-rated health (S2 Fig), and when start of follow-up was postponed three years (S3 Fig). When only the underlying cause of death was considered, the risk

**Table 1. Participant characteristics by smoking status in 1997: Never, former or current.**

|  | Smoking | | |
|---|---|---|---|
|  | **Never** | **Former** | **Current** |
| No. | 26,377 (41.9) | 21,405 (34.0) | 15,121 (24.0) |
| Age, years (IQR) | 61.0 (54.0–69.0) | 59.0 (53.0–68.0) | 58.0 (52.0–66.0) |
| Alcohol, g/day (IQR) | 5.6 (1.3–13.7) | 11.9 (4.5–23.2) | 10.8 (3.7–22.9) |
| Pack years (IQR) | 0 | 13.0 (6.3–22.8) | 21.6 (11.0–32.0) |
| Latest reported cigarettes/day (IQR) | 0 | 12.0 (8.0–20.0) | 10.0 (7.0–20.0) |
| Sex, n (%) |  |  |  |
| Men | 12,900 (48.9) | 14,880 (69.5) | 8,985 (59.4) |
| Women | 13,477 (51.1) | 6,525 (30.5) | 6,136 (40.6) |
| Education, n (%) |  |  |  |
| <9 years | 9,615 (36.5) | 7,242 (33.8) | 5,757 (38.1) |
| 9–12 years | 1,735 (6.6) | 1,483 (6.9) | 958 (6.3) |
| ≥12 years | 5,148 (19.5) | 3,793 (17.7) | 2,321 (15.3) |
| Other | 9,879 (37.5) | 8,887 (41.5) | 6,085 (40.2) |
| Marital status, n (%) |  |  |  |
| Cohabiting | 21,285 (80.7) | 17,857 (83.4) | 11,553 (76.4) |
| Living alone | 5,092 (19.3) | 3,548 (16.6) | 3,568 (23.6) |
| Charlson's weighted comorbidity index, n (%) |  |  |  |
| 0 | 22,506 (85.3) | 17,432 (81.4) | 12,443 (82.3) |
| 1 | 2,309 (8.8) | 2,487 (11.6) | 1,661 (11.0) |
| 2 | 1,214 (4.6) | 1,047 (4.9) | 706 (4.7) |
| ≥3 | 348 (1.3) | 439 (2.1) | 311 (2.1) |
| Exercise, n (%) |  |  |  |
| <1 hour/week | 4,907 (18.6) | 4,124 (19.3) | 4,136 (27.4) |
| 1 hour/week | 5,589 (21.2) | 4,293 (20.1) | 3,291 (21.8) |
| 2–3 hours/week | 9,044 (34.3) | 7,010 (32.7) | 4,268 (28.2) |
| 4–5 hours/week | 3,416 (13.0) | 2,796 (13.1) | 1,563 (10.3) |
| >5 hours/week | 3,421 (13.0) | 3,182 (14.9) | 1,863 (12.3) |
| Walking/bicycling, n (%) |  |  |  |
| Hardly ever | 2,629 (10.0) | 2,641 (12.3) | 2,439 (16.1) |
| <20 minutes/day | 5,553 (21.1) | 4,799 (22.4) | 3,631 (24.0) |
| 20–40 minutes/day | 8,844 (33.5) | 6,612 (30.9) | 4,391 (29.0) |
| 40–60 minutes/day | 4,719 (17.9) | 3,616 (16.9) | 2,198 (14.5) |
| >60 minutes/day | 4,632 (17.6) | 3,737 (17.5) | 2462 (16.3) |

was higher in current smokers (S4 Fig). Adjustment for BMI (S5 Fig) and time after 1998 (S6 Fig) showed similar results as the main analysis.

## Pneumonia

In the analysis of pneumonia, 62,902 individuals contributed 1,400,955 PYAR, and 10,471 individuals suffered at least one episode of pneumonia, 383 individuals were admitted to an ICU due to pneumonia and 2,851 died from pneumonia. Compared to never smokers, former smokers had HR 1.17 (95% CI 1.12–1.23) of contracting pneumonia, HR 1.42 (95% CI 1.09–1.84) of ICU admission and HR 1.16 (95% CI 1.06–1.27) of dying in pneumonia. Compared to never smokers, current smokers had HR 1.42 (95% CI 1.35–1.49) of contracting pneumonia, HR 2.26 (95% CI 1.75–2.94) of ICU admission and HR 1.70 (95% CI 1.55–1.87) of dying in

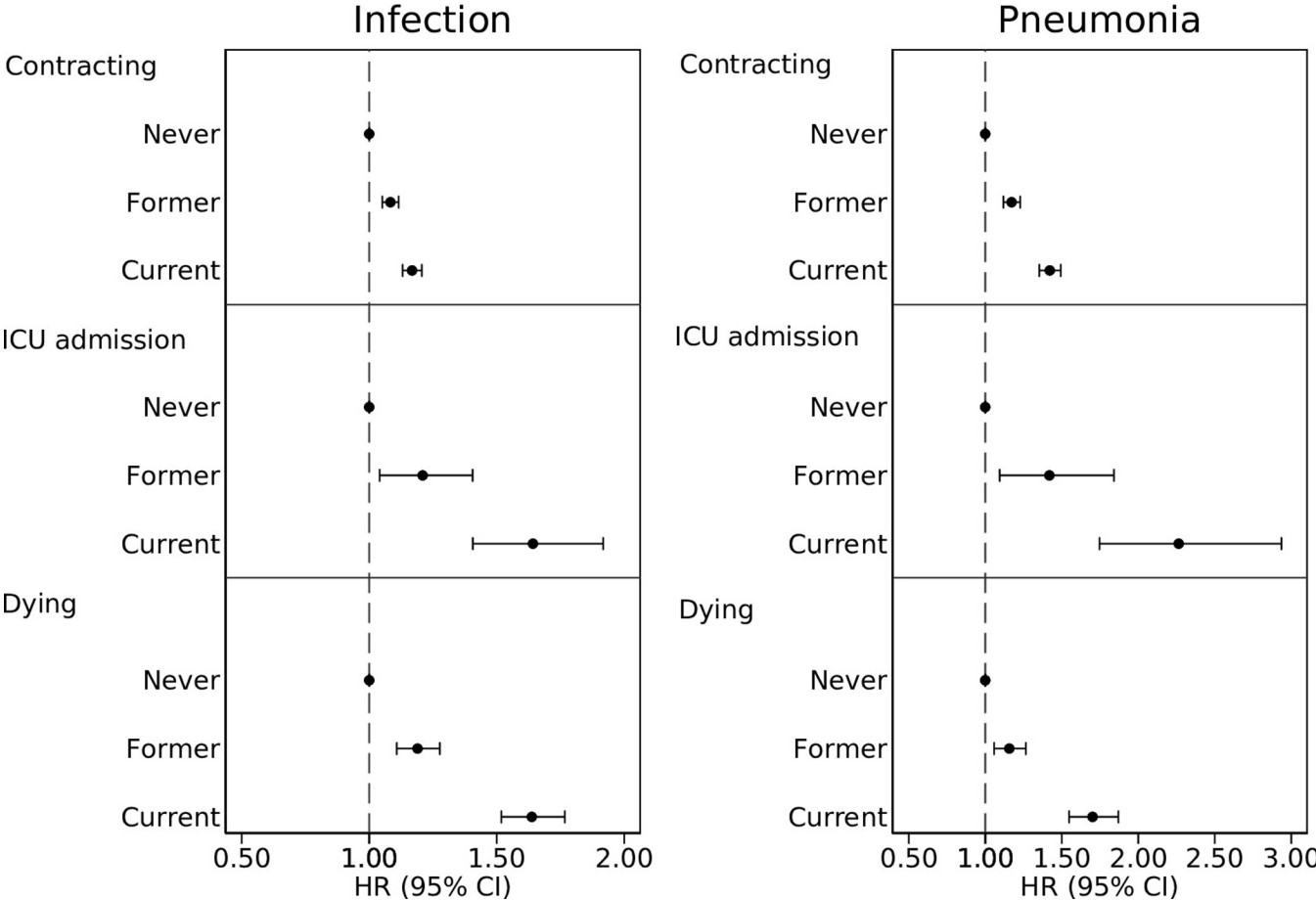

**Fig 1. Risk of contracting any infection and pneumonia, being admitted to and intensive care unit and dying by smoking status.** Hazard ratio (HR) and 95% confidence interval (CI) of contracting any infection or pneumonia; being admitted to an intensive care unit and dying, adjusted for age (as time scale), sex, Charlson's weighted comorbidity index, education, marital status, exercise, walking and alcohol consumption.

pneumonia (Fig 1 and S2 Table). Pack years smoked was positively associated with contracting pneumonia, ICU admission and dying in pneumonia, demonstrating a dose-response pattern. Years since smoking cessation had a dose-response negative association with the risk of contracting pneumonia, ICU admission and dying (Fig 2). Sensitivity analyses adjusted for self-rated health (S2 Fig) and when individuals became at risk three years after baseline (S3 Fig) showed similar results as the main analysis. In analysis of underlying cause of death, current smokers had higher risk of dying (S4 Fig). Analyses adjusted for BMI (S5 Fig) and time after 1998 (S6 Fig) showed similar, but attenuated, results as the main analysis. Former and current smokers had higher 30-day mortality after the first episode of pneumonia (S7 Fig).

## Discussion

In this study investigating two large representative population-based Swedish cohorts, compared to never smoking, former and current smoking at the baseline survey was associated with a higher risk of contracting any infection and pneumonia, as well as a higher risks of intensive care admission and death during follow-up. The risk of contracting and dying in infection and pneumonia increased with escalating number of pack years smoked and was negatively associated with years since smoking cessation. This does not support previous

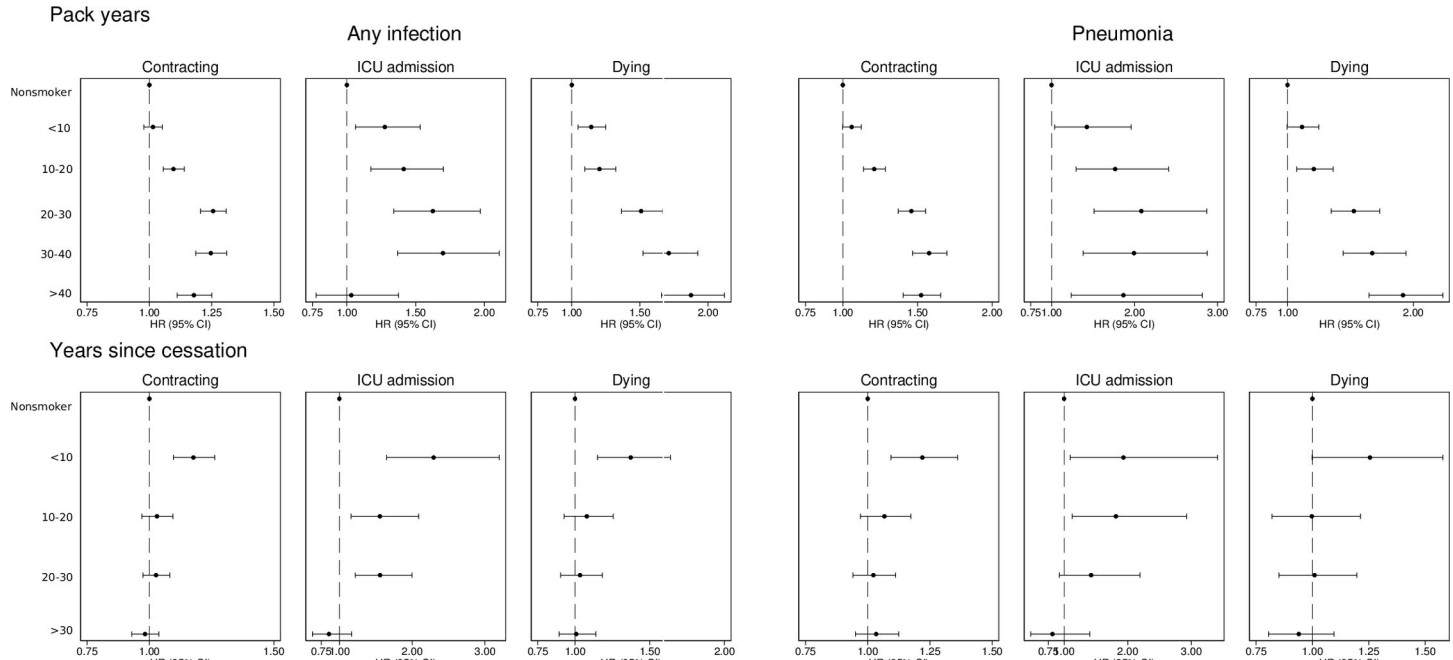

**Fig 2. Risk of contracting any infection and pneumonia, being admitted to and intensive care unit and dying by pack years and years since smoking cessation.** Hazard ratio (HR) and 95% confidence interval (CI) of contracting any infection or pneumonia; being admitted to an intensive care unit and dying for pack years (panel A) and years since smoking cessation (panel B). Adjusted for age (as time scale), sex, Charlson's weighted comorbidity index, education, marital status, exercise, walking and alcohol consumption.

findings where smoking was associated with a lower risk of dying in pneumonia [15–17], and highlights the importance of both primary smoking prevention and smoking cessation.

Although most previous research has focused on pneumonia, smoking has been associated with higher risk of other infections, such as abscess formation [31, 32], surgical site infection [33] and meningococcal disease [18]. The present study shows a generally increased risk of infection, as well as a higher risk of intensive care unit admission and risk of dying in infection. Although most previous studies have been unable to find an association between smoking and mortality in infection and sepsis [19–21], Huttunen et al [22] and a meta-analysis including 2,694 participants [23] found a higher risk of death in sepsis among smokers compared to non-smokers, similar to our findings.

The finding that both current and former smoking at baseline is associated with contracting pneumonia is supported by several studies [7–9] as are the opposing dose-response associations with pack years and years since smoking cessation [10]. Whether smoking increases the risk of dying in pneumonia is more controversial. Some studies have found a higher risk of death in pneumonia in smokers [12], whereas others show no [13, 14] or very weak [11] associations. Several studies investigating hospital-admitted patients have even shown *lower* risk of death among current smokers [15–17]. This disagreement with our results may be due to differences in method: although smokers may have a lower risk of dying during one specific hospitalisation; we find that they have a higher risk of intensive care unit admission and dying in pneumonia over time. Furthermore, our sensitivity analyses showed higher 30-day mortality after the first episode of pneumonia in former and current smokers, supporting our main analysis. We also found dose-response associations between contracting and dying in pneumonia with increasing pack years and recent smoking cessation. Indeed, the association between smoking and death in pneumonia was stronger than its association with contracting

pneumonia. Another point of contention is whether current smoking is associated with higher risk of dying in pneumonia compared to former smoking. Previous results are diverging, Bello *et al* found higher risk of dying in pneumonia among current smokers [12], whereas Almirall *et al* and Katanoda *et al* showed similar risks in former and current smokers [7, 11] and Marrie *et al* found higher risk of dying in former smokers [16]. We found that current smokers had the highest risk.

Strengths of this study include use of two large population-based, well-characterised cohorts representative of the underlying population followed for a very long time. Exposures were collected from detailed questionnaires and outcomes were ascertained from near-complete national registers with high reliability [28]. However, smoking status was ascertained in 1997 and could have changed subsequently. Further weaknesses include the lack of microbiological culture data, which may have yielded additional insights. Although analyses were adjusted for many confounders, the potential for residual confounding can never be excluded in observational studies. Smoking status was ascertained through self-report, which may underestimate the true prevalence of smoking as individuals tend to under-report socially undesirable behaviors [34], however this would bias any results towards the null. The Swedish Intensive Care Register started in 2001 and coverage of Swedish ICUs gradually increased over time [30]. It is therefore possible that episodes of ICU care early in the study period. Although the cohorts are representative of the Swedish population, results may not be applicable to other populations.

In conclusion, we found that smoking was associated with higher risk of contracting any infection and pneumonia, as well as higher risks of intensive care unit admission and death. Both former and current smoking at baseline were associated with elevated risks, emphasizing the importance of primary smoking prevention and smoking cessation.

## Supporting information

**S1 Table. Characteristics of participants admitted to an intensive care unit by smoking status.**
(XLSX)

**S2 Table. Regression output from analyses of smoking status.** Number of participants, number of cases, person-years at risk (PYAR), hazard ratio (HR) and 95% confidence interval (CI) for risk of contracting any infection or pneumonia, being admitted to an intensive care unit and dying. Crude model adjusted for age (as time scale) and sex. Adjusted model adjusted for age (as time scale), sex, Charlson's weighted comorbidity index, education, marital status, exercise, walking and alcohol consumption.
(XLSX)

**S1 Fig. Cohort flowchart.**
(TIF)

**S2 Fig. Sensitivity analysis adjusting for self-rated health.** Hazard ratio (HR) and 95% confidence interval (CI) of contracting any infection or pneumonia, being admitted to an intensive care unit and dying, adjusted for age (as time scale), sex, Charlson's weighted comorbidity index, education, marital status, exercise, walking, alcohol consumption and self-rated health.
(TIF)

**S3 Fig. Sensitivity analysis with follow-up beginning three years after baseline.** Hazard ratio (HR) and 95% confidence interval (CI) of contracting any infection or pneumonia, being admitted to an intensive care unit and dying with follow-up beginning three years after

baseline, adjusted for age (as time scale), sex, Charlson's weighted comorbidity index, education, marital status, exercise, walking and alcohol consumption.
(TIF)

**S4 Fig. Sensitivity analysis only considering the underlying cause of death.** Hazard ratio (HR) and 95% confidence interval (CI) of dying in infection and pneumonia, where only the underlying cause of death is considered, adjusted for age (as time scale), sex, Charlson's weighted comorbidity index, education, marital status, exercise, walking, and alcohol consumption.
(TIF)

**S5 Fig. Sensitivity analysis adjusting for body mass index.** Hazard ratio (HR) and 95% confidence interval (CI) of contracting any infection or pneumonia; being admitted to an intensive care unit and dying, adjusted for age (as time scale), sex, Charlson's weighted comorbidity index, education, marital status, exercise, walking, alcohol consumption and body mass index.
(TIF)

**S6 Fig. Sensitivity analysis with time from baseline as time scale.** Hazard ratio (HR) and 95% confidence interval (CI) of contracting any infection or pneumonia; being admitted to an intensive care unit and dying, adjusted for time from baseline (as time scale), sex, Charlson's weighted comorbidity index, education, marital status, exercise, walking, alcohol consumption and age.
(TIF)

**S7 Fig. 30-Day mortality after first diagnosis of pneumonia.** Hazard ratio (HR) and 95% confidence interval (CI) of 30-day mortality after the first diagnosis of pneumonia, adjusted for age (as time scale), sex, Charlson's weighted comorbidity index, education, marital status, exercise, walking, and alcohol consumption.
(TIF)

## Acknowledgments

We acknowledge SIMPLER (Swedish Infrastructure for Medical Population-Based Life-Course and Environmental Research) for provisioning of facilities and we would like to thank Anna-Karin Kolseth and Niclas Håkansson for assistance. The computations were enabled by resources in project SIMP2023020 provided by the National Academic Infrastructure for Supercomputing in Sweden (NAISS) at Uppsala Multidisciplinary Center for Advanced Computational Science (UPPMAX). We acknowledge all participating ICUs in the Swedish Intensive Care Registry for their participation and hard work to contribute data.

## Author Contributions

**Conceptualization:** Karl Stattin.

**Data curation:** Karl Stattin.

**Formal analysis:** Karl Stattin.

**Methodology:** Karl Stattin, Mikael Eriksson, Robert Frithiof, Rafael Kawati, Douglas Crockett, Michael Hultström, Miklos Lipcsey.

**Supervision:** Miklos Lipcsey.

**Visualization:** Karl Stattin.

**Writing – original draft:** Karl Stattin.

**Writing – review & editing:** Karl Stattin, Mikael Eriksson, Robert Frithiof, Rafael Kawati, Douglas Crockett, Michael Hultström, Miklos Lipcsey.

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
