## [Decision Letter · Decision Letter 0]

1 Apr 2024

PONE-D-24-06732Current and former smoking is associated with higher risk of contracting bacterial infection and pneumonia, intensive care unit admission and deathPLOS ONE

Dear Dr. Stattin,

Thank you for submitting your manuscript to PLOS ONE. After careful consideration, we feel that it has merit but does not fully meet PLOS ONE’s publication criteria as it currently stands. Therefore, we invite you to submit a revised version of the manuscript that addresses the points raised during the review process.

Dear Authors,

Thanks for the interesting conduct of the two cohorts follow up study on smoking status and infections incidence, ICU admission and death.

As you clearly state within the limitations paragraph, smoking status was ascertained at the beginning of the period and status could have changed over the period; a more worrying aspect to me is that authors state ( even in the title) that "Current and former smoking is associated with higher risk of contracting bacterial infection and pneumonia, intensive care unit admission and death"; if we are strict, what moment does current smoking refer to? I guess that to the moment when the outcome happens, is it correct? If death or pneumonia or any infection or ICU admission happens, this event is analyzed according to self declared smoking status at the beginning of the period, hence the word "current smoker" does not apply to an event taking place, let us imagine, in 2005 and smoking status has been asked in 1997.

If authors manage to display their results without falling into this inaccuracy in the way the express the exposure and the results, I would recommend to publish this piece of research.

Please take into account also the reviewers comments.

We look forward to receiving your revised manuscript.

Kind regards,

Ignacio Garitano-Gutierrez, M.D., MSc., EPIET.

Academic Editor

PLOS ONE

Journal Requirements:

Additional Editor Comments:

Dear Authors,

Thanks for the interesting conduct of the two cohorts follow up study on smoking status and infections incidence, ICU admission and death.

As you clearly state within the limitations paragraph, smoking status was ascertained at the beginning of the period and status could have changed over the period; a more worrying aspect to me is that authors state ( even in the title) that "Current and former smoking is associated with higher risk of contracting bacterial infection and pneumonia, intensive care unit admission and death"; if we are strict, what moment does current smoking refer to? I guess that to the moment when the outcome happens, is it correct? If death or pneumonia or any infection or ICU admission happens, this event is analyzed according to self declared smoking status at the beginning of the period, hence the word "current smoker" does not apply to an event taking place, let us imagine, in 2005 and smoking status has been asked in 1997.

If authors manage to display their results without falling in this inaccuracy in the way the express the exposure and the results, I would recommend to publish this piece of research.

Please take into account also the reviewers comments.

Reviewers' comments:

Reviewer's Responses to Questions

**Comments to the Author**

1. Is the manuscript technically sound, and do the data support the conclusions?

Reviewer #1: Yes

Reviewer #2: Yes

2. Has the statistical analysis been performed appropriately and rigorously? 

Reviewer #1: Yes

Reviewer #2: Yes

3. Have the authors made all data underlying the findings in their manuscript fully available?

Reviewer #1: Yes

Reviewer #2: Yes

4. Is the manuscript presented in an intelligible fashion and written in standard English?

Reviewer #1: Yes

Reviewer #2: Yes

5. Review Comments to the Author

Reviewer #1: The manuscript clearly describes the origin of the data collected, with big sample record achieved from two strong national databases, representative of the population they intended to study; the size is appropiate, meeting statistically significant results. The statistical analysis is well explained, following the main objectives of the research. Differences between the results of previous publications on the topic have been described, explaining the source of disagreements. The conclusion is supported by the results shown at the manuscript.

Reviewer #2: Congratulations for the work done. This is a very interesting study, which investigates the influence of current and previous tobacco consumption on the development and clinical course of respiratory infection and pneumonia.

The article deserves to be published, and I would only make two suggestions for modification:

On the one hand, in the abstract, it appears that the risk of admission to ICU is analyzed. This point is met in the full article, but not in the abstract, so I think it should be removed from the abstract, or include this data in it.

On the other hand, this study is carried out in a sample that represents the Swedish population. It may be that in other populations this same pattern does not hold true, and it should be mentioned among the limitations of the study.

With these two small changes, the article could be published.

6. PLOS authors have the option to publish the peer review history of their article (what does this mean?). If published, this will include your full peer review and any attached files.

Reviewer #1: **Yes: **Diego Manzano Moratinos

Reviewer #2: No

---

## [Author Response · Author response to Decision Letter 0]

4 Apr 2024

Dear Editor,

We would like to thank the Editor for the opportunity to improve our manuscript and the Reviewers for their kind and insightful comments. We have made adjustments throughout the manuscript in accordance with the suggestions, which we hope are satisfactory. Please see attached manuscript with changes highlighted. 

Sincerely, 

The authors, through Karl Stattin

Dear Authors,

Thanks for the interesting conduct of the two cohorts follow up study on smoking status and infections incidence, ICU admission and death.

As you clearly state within the limitations paragraph, smoking status was ascertained at the beginning of the period and status could have changed over the period; a more worrying aspect to me is that authors state ( even in the title) that "Current and former smoking is associated with higher risk of contracting bacterial infection and pneumonia, intensive care unit admission and death"; if we are strict, what moment does current smoking refer to? I guess that to the moment when the outcome happens, is it correct? If death or pneumonia or any infection or ICU admission happens, this event is analyzed according to self declared smoking status at the beginning of the period, hence the word "current smoker" does not apply to an event taking place, let us imagine, in 2005 and smoking status has been asked in 1997.

If authors manage to display their results without falling into this inaccuracy in the way the express the exposure and the results, I would recommend to publish this piece of research.

Please take into account also the reviewers comments.

We agree. The wording is problematic as smoking status may indeed have changed during follow-up and it is unclear what timepoint “current” refers to. The title has been changed to "Smoking is associated with higher risk of contracting bacterial infection and pneumonia, intensive care unit admission and death", and the manuscript has been altered throughout to reflect that "current smoking" refers to ongoing smoking at baseline in 1997.

Reviewer #1: The manuscript clearly describes the origin of the data collected, with big sample record achieved from two strong national databases, representative of the population they intended to study; the size is appropiate, meeting statistically significant results. The statistical analysis is well explained, following the main objectives of the research. Differences between the results of previous publications on the topic have been described, explaining the source of disagreements. The conclusion is supported by the results shown at the manuscript.

Thank you for your kind words.

Reviewer #2: Congratulations for the work done. This is a very interesting study, which investigates the influence of current and previous tobacco consumption on the development and clinical course of respiratory infection and pneumonia.

The article deserves to be published, and I would only make two suggestions for modification:

On the one hand, in the abstract, it appears that the risk of admission to ICU is analyzed. This point is met in the full article, but not in the abstract, so I think it should be removed from the abstract, or include this data in it.

Good point. The abstract is unfortunately already at the word limit. “ICU admission” is thus removed from the aims in the abstract to improve consistency. 

On the other hand, this study is carried out in a sample that represents the Swedish population. It may be that in other populations this same pattern does not hold true, and it should be mentioned among the limitations of the study.

With these two small changes, the article could be published.

Indeed. This has been added as a limitation to the discussion section: “Although the cohorts are representative of the Swedish population, results may not be applicable to other populations.”

---

## [Editor Report · Decision Letter 1]

8 Apr 2024

Smoking is associated with higher risk of contracting bacterial infection and pneumonia, intensive care unit admission and death

PONE-D-24-06732R1

Dear Dr. Stattin,

We’re pleased to inform you that your manuscript has been judged scientifically suitable for publication and will be formally accepted for publication once it meets all outstanding technical requirements.

Kind regards,

Ignacio Garitano-Gutierrez, M.D., MSc., EPIET.

Academic Editor

PLOS ONE